# Targeting Gametocytes of the Malaria Parasite *Plasmodium falciparum* in a Functional Genomics Era: Next Steps

**DOI:** 10.3390/pathogens10030346

**Published:** 2021-03-16

**Authors:** Jyotsna Chawla, Jenna Oberstaller, John H. Adams

**Affiliations:** 1Molecular Medicine, Morsani College of Medicine, University of South Florida, 12901 Bruce B Downs Blvd, MDC 7, Tampa, FL 33612, USA; jyotsna1@usf.edu; 2Center for Global Health and Infectious Diseases Research and USF Genomics Program, College of Public Health, University of South Florida, 3720 Spectrum Blvd, Suite 404, Tampa, FL 33612, USA; jobersta@usf.edu

**Keywords:** gametocyte biology, sexual stages, forward genetic screens, transmission-blocking candidates

## Abstract

Mosquito transmission of the deadly malaria parasite *Plasmodium falciparum* is mediated by mature sexual forms (gametocytes). Circulating in the vertebrate host, relatively few intraerythrocytic gametocytes are picked up during a bloodmeal to continue sexual development in the mosquito vector. Human-to-vector transmission thus represents an infection bottleneck in the parasite’s life cycle for therapeutic interventions to prevent malaria. Even though recent progress has been made in the identification of genetic factors linked to gametocytogenesis, a plethora of genes essential for sexual-stage development are yet to be unraveled. In this review, we revisit *P. falciparum* transmission biology by discussing targetable features of gametocytes and provide a perspective on a forward-genetic approach for identification of novel transmission-blocking candidates in the future.

## 1. Introduction

Malaria is a devastating vector-borne disease that threatens the health of about half of the world’s population. It is caused by obligate intracellular parasites of the *Plasmodium* genus and is transmitted by the bite of an infected *Anopheles* mosquito vector. Of the five *Plasmodium* species that cause infection in humans, *Plasmodium falciparum* is responsible for highest mortality [1,2]. The Sub-Saharan African region accounts for more than 90% of global malaria deaths, disproportionately comprised of pregnant women and young children (<5 years of age) [1]. The eukaryotic parasite has a complex life cycle and is primed for survival in its human intermediate host and mosquito definitive host. Human infection begins when *Plasmodium* sporozoites are injected into the skin and migrate to the liver. Following a silent asexual replication in the liver during a 10-day incubation period, erythrocytic-invasive forms known as merozoites are released into the bloodstream. *P. falciparum* cell tropism during the erythrocytic stage is not limited to reticulocytes, leading to increased parasitemia levels in a shortened timespan [3]. Ensuing parasite population explosions occur every 48 h, corresponding to the length of an asexual or erythrocytic cycle and resulting in the classic clinical manifestations of paroxysmal fever, chills, myalgia, vomiting, and severe anemia. The major pathophysiological processes in *P. falciparum* malaria are due to the parasitized red blood cells’ (pRBCs) cytoadhesion to the endothelium, platelets and uninfected erythrocytes (rosetting), leading to sequestration of pRBCs that block microvasculature and cause complications such as cerebral malaria, stroke and death [4,5]. As the parasite burden increases, malaria parasites face an important decision of whether to continue proliferation or invest in sexual transmission, which is a developmental dead-end if not ingested by a susceptible anopheline mosquito. A developmental switch occurs in a proportion of the asexually replicating parasites, causing them to exit the asexual phase and commit to sexual development. Sexual differentiation gives rise to transmission stages known as gametocytes that are solely responsible for transmitting parasites to the mosquito vector. This stage makes for a promising point of transmission intervention, as very few parasites differentiate into gametocytes and even fewer are permitted to develop in the vector [6,7,8]. Though gametocytes were first described in 1880 by Dr. Alphonse Laveran, critical gaps in our understanding of gametocyte biology still remain, largely due to the technical difficulties associated with in vitro culturing and the challenges related to in vivo tracking of low-density gametocytes that tend to go undetected by routine microscopy [9,10].

Targeting gametocytes will not only prevent the spread of malaria through the local community, but also reduce the overall population disease burden in tropical regions. Historically, public health interventions that seek to disrupt mosquito transmission through environmental elimination or aim to limit human exposure to mosquito vectors have been very effective in reducing malaria prevalence in many but not all endemic regions, if sustained. The current goal to eradicate malaria globally, established by the Malaria Eradication Research Agenda (malERA) program achieved considerable success with the use of insecticide-treated mosquito bed nets (ITNs) and artemisinin combination therapies (ACTs), decreasing malaria cases and mortality rate even in high-transmission areas [11]. Unfortunately, the past decade’s remarkable progress in malaria elimination is currently being threatened by emerging drug resistance of the parasite to the frontline drug artemisinin and partner-drugs administered in ACT, as well as reduced effectiveness of ITNs [12,13]. Mutations in the *PfKelch13* propeller domain, related to artemisinin resistance, have emerged in South-East Asia, with a few early signs of appearance in some regions of Africa [14,15]. Insecticide resistance to pyrethroids has also been extensively observed in Sub-Saharan Africa [16,17]. In the past, *P. falciparum* has shown resistance to almost all previous leading antimalarial drugs [18,19,20,21], and complete artemisinin resistance may be inevitable. In recent years, we have seen a renewed interest in transmission-blocking interventions to help contain the spread of malaria as a necessary component of the elimination strategy. Core tenets of a holistic transmission-interruption strategy would include: (i) gametocidal drugs, to clear gametocytes from the circulation and render hosts non-infectious to mosquitoes; (ii) transmission-blocking vaccines, to induce immunity in humans to prevent further development of transmissible forms in the mosquito; and (iii) vector-control, to shift the equilibrium towards refractory mosquitoes [22,23]. A comprehensive understanding of transmission biology is critical for achieving the herculean goal of effective disease control and eradication. In this review, we highlight critical factors of *P. falciparum* gametocyte biology, the current state of antimalarial drugs and vaccines in relation to transmission and the importance of functional genomic approaches in the quest for new transmission-blocking candidates.

## 2. Understanding *P. falciparum* Gametocyte Biology

Inside the vertebrate host, the parasite’s course of infection can be divided into three parts: (i) pre-erythrocytic phase that begins with the migration of sporozoites from the site of injection to the liver, followed by an incubation period in which the parasites divide asexually to produce invasive forms called merozoites; (ii) erythrocytic phase that is characterized by merozoite invasion of red blood cells and multiple rounds of asexual reproduction corresponding to the clinical symptoms; (iii) sexual differentiation phase that sets in with the expression of commitment marker genes. Sexual differentiation is an obligate step for parasite survival and transmission and the decision to commit to sexual development is taken during active infection under the influence of signals from the host (physiological stress) and parasites (autocrine factors) [24]. Commitment is brought upon by derepression of an epigenetically silenced *ap2-g* gene locus in replicating parasites [25,26,27]. The activation results in sexually committed schizonts producing sexually committed daughter merozoites destined to differentiate into mature gametocytes upon reinvasion [28]. Gametocyte conversion can also occur earlier at the ring-stage, leading to direct differentiation without the need of an asexual division cycle [29].

In comparison to the 24–60 h required for intraerythrocytic gametocyte maturation by other *Plasmodium* species, *P. falciparum* gametocytes require an extended period of 8–12 days for generation of transmission-competent forms [30,31]. A series of morphological changes accompany *P. falciparum* gametocyte development that can be visibly distinguished into five stages as described by light microscopy and ultrastructural analyses [32,33]. Gametocyte stages I-IV sequester in the bone marrow while stages V are released into circulation where they remain for several days (3.5–6.5 days) [34]. Stage V gametocytes are considered mature when they become infectious for the mosquito vector and accumulate in the microvasculature of the dermis to facilitate mosquito ingestion [35,36]. These mosquito-infective stages are easily identifiable by a characteristic crescent or falciform shape unique to *P. falciparum*, a feature that is considered to be retained from its evolutionarily close relative of avian malaria parasites [37,38,39]. The peripheral gametocyte density is usually around ~100/μL [40], though there have been reports of successful transmissions even at submicroscopic densities [41,42]. *Plasmodium* gametocytes are dimorphic—all merozoites from a single schizont develop into either microgametocytes (male gametocytes) or macrogametocytes (female gametocytes) [43]. The sexes can be differentiated via routine Giemsa-stained blood film, as males appear pink with blunt edges, and usually have a scattered malaria pigment, while the females stain blue with angular edges and a dense nucleus [44]. The primary function of a male gametocyte is to produce 8 male sperm-like gametes to fertilize macrogametes [45,46,47]. The female gametocyte on the other hand, is prepared for fertilization and continued development in the mosquito vector. Gametocytes are highly specialized precursor cells that produce gametes on acquiring cues from their new environment in the mosquito vector. Upon egress from the RBCs in the shelter of blood bolus, haploid male and female gametes fuse to form a zygote, the only diploid stage in parasite development. Following recombination and chromosomal segregation, the zygote progresses into motile ookinetes. Ookinetes penetrate the mosquito midgut epithelium and develop into round sessile oocysts to establish infection. A few days later oocysts rupture to release sporozoites that migrate to the salivary glands, where they remain capacitated and ready to infect when the mosquito takes a bloodmeal. Gametocytes thus provide the essential link for human-to-vector transmission and serve as a promising target to interrupt the parasite life cycle and decrease malaria prevalence. Discussed below are striking features of *P. falciparum* gametocyte biology that can be exploited to advance novel transmission-blocking strategies.

### 2.1. Gametocyte Conversion

Gametocyte conversion can be defined as the rate at which sexual commitment occurs in a given asexual parasite population at a given time. Usually only a small proportion (0.1–5%) of asexual parasites develop into gametocytes during each cycle [48,49]. The conversion rates vary between isolates and clones of the same isolates [50]. Initially, gametocytogenesis was observed as a parasite response to stress in the human host. Efforts to decode this stress signal over the years have led to the identification of many factors that influence conversion, including host environmental factors, antimalarial drug treatments and a few others [51,52]. Both naturally acquired humoral and non-specific cellular immunity in malaria influence sexual stages [53]. An increase in gametocyte production in vitro has been observed on addition of sera from malaria-infected children and with treatment of anti-*P. falciparum* antibodies produced by hybridoma cell lines [54,55]. Epidemiological studies show a negative correlation between gametocyte peaks and hemoglobin concentration [56], suggesting an increased gametocyte carriage in anemic population [57]. In vitro culturing of *P. falciparum* gametocytes has further helped gain insight into the effects of the parasite’s environment on sexual conversion. An important finding has been an increase in gametocyte conversion on depletion of host-derived lipid lysophosphatidlycholine (LysoPC) [58]. In addition, the presence of extracellular vesicles (EV) secreted by the parasites in the conditioned medium serves as a signal to induce gametocytogenesis [59]. Some antimalarials have also been shown to promote sexual development in the host (refer to antimalarial drug therapies and gametocytes section). Recently, parasite factors like genetic determinants driving sexual development have been identified with the discovery of AP2-G as a master regulator of sexual commitment in *P. falciparum* [25]. Two other proteins histone deacetylase 2 (PfHda2) [60] and heterochromatin protein 1 (PfHP1) [61,62], epigenetically regulate AP2-G, preventing its expression in asexual parasites. Another protein, gametocyte development 1 (GDV1), releases derepression and activates *ap2-g* by eviction of HP1 [63]. Recent studies reported that a truncation of GDV1 C-terminal results in the disruption of sexual commitment [64]. Interestingly, expression of this master regulator is controlled by a long non-coding antisense RNA [63]. The upstream factors regulating sexual conversion still remain unknown making it essential to decipher molecular mechanisms underlying sexual commitment. Conversion is the first step towards transmission and interrupting this stage of the parasite can go a long in way in malaria elimination.

### 2.2. Gametocyte Sequestration

Sequestration of pRBCs, infected with asexual stage trophozoites and schizonts from circulation to avoid clearance by spleen, have been associated with severe disease pathology of *P. falciparum*. Adherence to the host cell receptors is mediated by a knob-like protein *P. falciparum* erythrocyte membrane protein-1 (PfEMP-1) on the surface of infected erythrocytes [65,66]. Only recently is the discovery that *P. falciparum* immature gametocytes sequester in the bone marrow and spleen [34,67,68]. The bone marrow niche offers compelling reasons for sequestration, including the presence of erythroblastic islands [69,70], protection from immune clearance as a result of central tolerance [71] and the depletion of phospholipids [58], all of which have been positively implicated in gametocytogenesis. It is speculated that sexually committed merozoites and young gametocytes home to the bone marrow and alternatively committed-schizonts give rise to gametocytes that develop in the bone marrow parenchyma [72]. Gametocyte sequestration is pfEMP-1 independent, however low levels of pfEMP1 in early gametocyte stages could be responsible for weak initial interactions [73]. A proteomic analysis of early gametocytes identified a group of gametocyte-exported parasite proteins (GEXPs) [74]. Two gametocyte surface antigens, GEXP10 and GEXP7, were shown to interact with human chemokine fractalkine receptors (CX3CL1) present on bone marrow stromal cells [75,76]. Retention of gametocytes in the bone marrow could likely involve adhesion from potential GEXPs ligands and rheological properties like rigidity of gametocytes (I-IV). The bone marrow as a reservoir for gametocytes offers new possibilities to strategically hamper gametocyte development and interrupt transmission. Targeting the sequestration phenomena unique *to P. falciparum* gametocytes calls for further research directed towards the discovery of: (i) antigens for host cell invasion of sexual merozoites; (ii) antigens and receptors enabling extravasation into the bone marrow parenchyma; and (iii) interacting partners at the erythroblastic islands that favor gametocytogenesis.

### 2.3. Gametocyte Development and Maturation

The outcome of differential gene expression patterns is the production of proteins unique to gametocytes, one that encourages drastic host cell remodeling as well as visible changes in the parasite morphology fundamental to the sexual development process. Following invasion by a sexually committed merozoite, gametocytes grow to occupy much of the red blood cell volume over a 10-day period of molecular, metabolic and cellular rewiring.

#### 2.3.1. Molecular Biology

Gametocytes are terminally differentiated cells arrested in the G0 phase of the cell cycle and do not undergo DNA synthesis during their course of development until exflagellation (a Medusa-like activation of the microgametocyte to produce male gametes) [77,78,79]. Nucleic acid synthesis is thus restricted to RNA synthesis which is viewed to cease by day 6 in gametocytes [80,81,82]. The rates of protein turnover and hemoglobin digestion also come to a halt, explaining the loss of sensitivity of late-stage gametocytes to the most commonly used antimalarial drugs. Despite reaching a quiescent stage of the life cycle, gametocytes contain a high proportion of untranslated sexual-stage specific mRNA transcripts [83,84]. Many of these mRNAs code for gamete proteins necessary for fertilization and advancement in the mosquito vector and are found in a translationally repressed state in gametocytes [85]. This strategy of post-transcriptional regulation allows the parasite to respond swiftly to signals from the new host environment in an event of transmission. Translational regulators like DOZI (development of zygote inhibited) and CITH (CAR-I and fly Trailer Hitch, worm homolog) are well characterized in the rodent-infective malaria model [86,87]. These mRNA binding proteins regulate the storage of mRNA transcripts in the form of ribonucleoprotein complexes in female gametocytes. In *P. falciparum*, Puf2 from the PUF-family of proteins binds to mRNA targets like P25 and P28 and the resulting disruption of the *puf2* gene enhances gametocytogenesis with an increase in number of male gametocytes [88,89]. *Plasmodium* expresses a large number of RNA-binding proteins, and the disruption of appropriate protein-RNA interactions and their trafficking can indeed serve as possible targets to arrest transmission [90,91].

#### 2.3.2. Metabolism

*Plasmodium falciparum* possesses two extrachromosomal genomes, a 35-kb apicoplast circular DNA and a 6-kb mitochondrial DNA element that are maternally inherited [92]. While the apicoplast remains morphologically static, the mitochondrion undergoes elaborate changes during sexual development. In contrast to the asexual-blood stages, the mitochondrion elongates and increases the tubular branching of its cristae in gametocytes [93], implying the role of mitochondria for energy production in sexual development. As evident from transcriptomic and other studies, an upregulation of *cytochrome b* gene and most of the mitochondrial enzymes indicate an active TCA cycle and increased glucose catabolism, making gametocytes susceptible to electron transport chain inhibitors [94,95]. Targeting mitochondrial metabolic pathways could generate defects in gametocytogenesis and disrupt transmission. Furthermore, gametocyte expansion and organelle enlargement during development increases the demand for cellular and membrane lipids. The gametocyte lipid repertoire differs greatly from asexual stages and also shows a variation between early and late sexual stages, indicating an active lipid metabolism throughout the gametocyte cycle [96,97]. As gametocytes mature, there is a general reduction in membrane phospholipids, especially lysophospholipids and fatty acid reservoirs like glycerolipids [96]. These findings are concordant with previous studies where depletion of lysophosphatidlycholine (LysoPC) has been linked to gametocytogenesis [58]. Meanwhile, lipidomic analysis reveals an enrichment of sphingolipids and ceramides in gametocytes. The increased requirement for *de novo* biosynthesis of ceramides has been confirmed by the sensitivity of gametocytes to ceramide inhibitors [96]. Whether these membrane-fluidity regulating lipids are playing a role in signaling pathways is yet to be determined. With growing evidence of the importance of lipid metabolism in sexual stages, uncovering key lipid enzymes through functional genomic studies is foreseeable. These enzymes have the potential to be emerging drug targets to prevent transmission by interfering with gametocyte development [98].

#### 2.3.3. Cellular Biology

Morphological transformation is a long-standing hallmark associated with *P. falciparum* sexual development and is crucial for transmissibility. Stage I gametocytes are indistinguishable from the asexual trophozoites, with changes becoming more apparent at stage II. The characteristic ‘D’ shaped stage II gametocytes elongate further and have blunt ends at stage III. While gametocytes assume a spindle shape at stage IV, they lose their rigidity to form crescent-shaped stage V gametocytes. These radical changes are possible due to the reversible expansion of cytoskeletal elements that allow gametocytes to make a transition from the site of sequestration to peripheral circulation [99,100]. About 10% of the gametocyte proteome is exported out into the host cell to mediate structural and mechanical remodeling of the erythrocyte membrane. Gametocyte export proteins (GEXPs) are expressed early on at the onset of gametocytogenesis and reach the host cytosol by passage through a parasite derived translocon known as PTEX (*Plasmodium* translocon of exported proteins) [74,101]. One such complex that drives gametocyte metamorphosis is the inner membrane complex (IMC) developing underneath the plasmalemma [102,103]. The IMC contains some gliding-associated proteins (GAP-50, GAP-45) that interact with the red blood cell spectrin-ankyrin network [104,105]. IMC formation is progressive, forming at one end of the gametocyte and extending to encircle the entire gametocyte followed by a collapse of the IMC to yield deformable gametocytes at maturation. A coordinated assembly of microtubule networks underneath the IMC drives the elongation process [102,103]. Proteins of the multigene STEVOR family are highly expressed in the gametocyte stages and play an important role of membrane stabilization during elongation. Dephosphorylation of STEVOR obliterates its contact with the ankyrin complex and aids in shape-shifting of the gametocytes [106,107]. Parasite cyclic-AMP (cAMP)-dependent kinase signaling regulates this ‘deformability switch’. A decrease in cAMP triggers deformation, while drugs that raise cAMP levels contribute to gametocyte stiffness, preventing its maturation [108]. *P. falciparum* has adopted an elegant mechanism of producing mature transmission-ready gametocytes. Functional genomics offer the potential to pursue genome-scale approaches to seek answers for the many outstanding questions about the mechanisms underlying the expansion of IMC, the dynamics of microtubule recruitment, the roles of interacting partners of the membrane complex, and the triggers of deformability for the identification of new transmission-blocking targets.

### 2.4. Gametocyte Sex Ratio and Exflagellation

The decision to form a male or female gametocyte is not determined by sex chromosomes but rather differential gene expression of the haploid genome is responsible for achieving different features of the two sexes [43,109,110]. In many ways, *P. falciparum* sexual development is homologous to the maintenance of sex in plants and algae, which in part can be explained by the existence of cytoplasmic incompatibility between the gametes and uniparental inheritance of organelles like the mitochondria and apicoplast [92,111]. In addition, strain-vector compatibility influences the chances of transmission, as certain mosquito-vector immune systems limit infections of particular parasite strains while remaining susceptive to infection by other strains [22]. Successful malaria transmission requires the presence of at least one male and one female gametocyte. Transcriptomic studies coupled with proteomic analysis revealed about half of the quantifiable proteome was differentially expressed in *P. falciparum* male and female gametocytes [112]. Proteins involved in DNA replication, motility, chromosome reorganization, and axoneme formation were found to be enriched in males whereas female-specific proteins were geared toward energy metabolism, translation and organellar functions [112]. ‘Omics’ analysis observed a sex-specific distribution of signaling proteins with an upregulation of five kinases and ten phosphatases in female gametocytes in contrast to ten kinases and three phosphatases upregulated in male gametocytes [112]. Sex-specific differences in surface proteins are used to quantify gametocytes in vivo as well as for in vitro analysis through PCR (qRT-PCR). *Pfs230* and *pfs25* were identified as the first male and female markers, respectively, to be used in diagnostic tools [113]. New improved versions of diagnostic assays that determine sex ratios use male markers such as *pf13* and *pfGMET* and female markers such as *pfGK* [114,115,116]. The sex ratio is the ratio of the number of male gametocytes to female gametocytes that varies in different carriers as well as during the course of infection. In *P. falciparum*, the sex ratio is female biased (three to four females to one male) to ensure fertilization as one microgametocyte releases eight male gametes while one macrogametocyte gives rise to only one female gamete [117]. The intricate balance between male and female gametocytes is critical to transmission and factors skewing this ratio have been of interest lately [118]. Host factors like anemia not only increase gametocyte production but also generate a male biased sex ratio with prolonged survival of male gametocytes to favor transmission [57,119,120]. Parasite factors like competition during multiclonal infections lead to a male biased sex ratio to maximize the chances of fertilization by its own genotype [121]. Clinical studies have also spotted a shift toward a male biased sex ratio due to delayed clearance of male gametocytes upon antimalarial drug treatment [122,123]. Some data suggest that male biased gametocyte sex ratios are more infectious to mosquitoes and are especially important for driving transmission at low gametocyte densities [117,121]. After the uptake of gametocytes by the mosquito vector, male gametocytes undergo exflagellation, a dramatic process of releasing gametes in response to signals such as rise in pH, drop in temperature and presence of xanthurenic acid in the mosquito’s midgut [124,125,126,127]. The genome is replicated three times in a short span of 15 min, making this replicative stage an attractive antimalarial target [78,128,129]. An increasing number of high-throughput drug screens include sex-specific responses for examining the effects on exflagellation, with a rationale of discovering new transmission-blocking molecules [130]. Sexual development in *P. falciparum* is a highly coordinated, complex process encompassing differential gene regulation, signaling and metabolic pathways that are yet to be deconvulated and validated by functional genomic approaches (Figure 1).

## 3. Antimalarial Drug Therapies and Gametocytes

Antimalarial therapy is indispensable to the control and prevention of malaria. We are in a constant battle with the apicomplexan parasite’s ability to adapt and gain resistance to antimalarial drugs [131,132], making it imperative to discover new druggable targets and to develop new therapies. Most of the leading antimalarial drugs are schizonticides, which target the asexually replicating parasites for management of symptoms in the host. A majority of these drugs are also effective against early gametocytes, as they work by blocking essential processes like RNA synthesis and hemoglobin digestion that are active in early stages (I, II, III) of gametocyte development [133]. There is little evidence on the impact of these drugs on later stages of gametocytes (IV, V) and transmission.

An 8-aminoquinoline primaquine (PQ) is the only approved antimalarial treatment that is effective against late-stage mature gametocytes. Morphological changes in the mitochondria have been associated with PQ mode of action [134,135]. A recent investigation demonstrated that hydroxylated-PQ metabolites were responsible for efficiently clearing out mature gametocytes by undergoing increased oxidation and accumulation of H_2_O_2_ in a two-step biochemical pathway [136]. The use of PQ to prevent transmission is limited due to significant concerns about the drug’s lethal hemolytic nature in individuals suffering from glucose-6-phosphate dehydrogenase (G6PD) deficiency, a genetic disorder that affects many people in malaria-endemic regions [137,138]. An analogue of PQ, bulaquine has shown higher efficiency in clearing gametocytes with reduced toxicity and has been approved for treatment in other *Plasmodium* species [139,140]. A synthetic antiparasitic agent, methylene blue (MB) serves as a promising alternative to primaquine by inhibiting all stages of gametocytes, including transmission to anopheline mosquitoes [141,142]. In a randomized phase II study, MB showed pronounced effects on existing and new gametocytes; however, more studies are needed to investigate the safety of MB, especially in patients with G6PD [143,144,145].

Treatment of *P. falciparum* malaria with the frontline ACTs show a decrease in gametocyte carriage and the subsequent number of infected mosquitoes [146,147]. Of the four ACTs recommended by WHO, the most widely utilized artemether–lumefantrine (AL) has been consistent in reducing gametocytemia in field studies [148,149]. This is thought to be a result of quick clearance of asexual parasites, which shortens the window for gametocyte formation. While the mechanism of action for artemisinin is not completely understood, there is evidence that free heme (a byproduct of the parasite’s digestion of host hemoglobin) acts on the endoperoxide bridge of artemisinin to generate volatile oxidizing products. These toxic products damage membrane proteins and interfere with normal organellar functions becoming more abundant as the parasite increases metabolic activity in later ring-stages of development, which can make artemisinins effective against early-stage gametocytes [150,151]. Another synthetic endoperoxide drug, artefenomel (previously known as OZ439), is currently under phase IIb clinical trials. OZ439 is a strong contender for a gametocidal drug as it seems to inhibit exflagellation and reduce the number of oocysts formation in vitro [133]. Like other artemisinin derivatives, artefenomel is thought to act by alkylating heme and inducing protein damage [152,153]. In a recent trial, a single dose of artefenomel administered as a monotherapy in combination with another antimalarial drug showed significant reduction in parasitemia and gametocytemia, supporting its use for treatment of *P*. *falciparum* malaria [154].

Commonly prescribed for malaria prophylaxis, atovaquone has shown potent gametocidal activity in vitro by clearing gametocytes through inhibition of the electron transport chain and parallel processes like pyrimidine synthesis [155,156,157]. Unfortunately, the solo use of this drug is not recommended because a single point mutation in the parasite’s *cytochrome b* gene can trigger drug resistance [158]. In vitro studies have confirmed the efficacy of atovaquone on parasite’s sexual and liver stages and current clinical trials are focused on testing new drug combinations with atovaquone-proguanil for the treatment of multidrug resistant parasites [159]. Another class of drugs that is currently being evaluated as antimalarial agents are proteasome inhibitors that affect asexual intraerythrocytic parasites by degrading intracellular proteins essential for parasite survival. The relevance of the proteasome in sexual stages of the parasite was demonstrated by the potent anti-gametocyte activity of a specific proteasome inhibitor, epoxomicin at nanomolar concentrations [160]. Interestingly, an antimicrobial drug thiostrepton that works by inhibiting protein synthesis displayed remarkable ability in eliminating asexual and sexual stages of *P. falciparum* by targeting the parasite proteasome and apicoplast [161]. A bioassay that tested 20 antimalarial drugs with a sex-specific readout reported increased sensitivity of male gametocytes to thiostrepton [130]. These studies suggest growing evidence of mitochondrial metabolism and proteasome pathways as emerging drug targets that hold the key for the development of novel transmission-blocking chemotherapeutic interventions [162,163,164]. Additionally, natural products from plant and microbial origins are being extensively investigated for the discovery of new antimalarial compounds [165].

A drug-treatment-induced increase in gametocyte density or prevalence has been an early parasitological indicator of emerging drug resistance, a phenomenon witnessed with chloroquine and sulfadoxine-pyrimethamine [166,167,168,169,170,171]. A recent study pointed out an enhanced transmission of artemisinin-resistant isolates in the Greater Mekong Subregion [172]. It is crucial to dissect the complex relationship between antimalarial drugs and gametocyte progression to effectively guide drug development and decrease the chances of drug-resistant parasite transmission. In the interest of malaria eradication, an ideal drug should reduce parasite burden and inhibit gametocytes to prevent the spread of malaria. To get to this point, we must probe for answers to remaining questions such as drug treatment and the competitive effect of multiple genotypes in co-infections, the impact on asynchronous gametocyte production that comes from varying conversion rates of different genotypes and the accessibility and pharmacokinetics of drugs for sequestered stages of gametocytes. A missing piece in this effort is the knowledge about genes encoding potential candidates and their role in transmission. A better understanding of gametocyte biology in terms of essential genetic determinants that direct and regulate the highly orchestrated process of sexual development is essential for drug discovery of new age antimalarials.

## 4. Transmission-Blocking Vaccines and Gametocytes

To reach the goal of malaria eradication, vaccines that interrupt transmission are sorely needed. The main premise of a transmission blocking vaccine (TBV) is the production of antibodies in the human host for an immune attack on sexual-stage antigens in the vector host [173]. One can be hopeful about the success of TBVs because sera from immune adults have been effective in inhibiting gamete fertilization and further development in the mosquito vector [174,175]. Naturally acquired transmission immunity occurs on exposure of proteins from disintegrated gametocytes that were left behind in circulation [176]. Ingestion of these antibodies during a bloodmeal blocks the function of gametes for the development of mosquito stages or initiates a complement mediated destruction of sexual stages [177,178]. These antigens are categorized as pre-fertilization antigens (Pfs230, Pfs48/45, Pfs47) and post-fertilization antigens (Pfs25, Pfs28) depending on their function in sexual development and are extensively described in other reviews [53,179]. A good TBV candidate is one that is less polymorphic, highly immunogenic and shows significant reduction in disease transmission. Two vaccine candidates, Pfs230, a 300 kDa belonging to the six-cysteine (6-cys) family of proteins [180,181] and Pfs25, a 25-kDa (glycosylphosphatidylinositol) GPI-anchored protein from the P25 family of proteins [182] are currently in phase I clinical trials. Expression of Pfs230 begins intracellularly in gametocytes while still in the human host, and the protein is essential for male fertility by forming a stable complex with Pfs48/45 [183,184]. The recombinant fragment of Pfs230 has shown promising results in two formulations: one conjugated to exoprotein A (EPA) and the other with aluminum hydroxide (Alhydrogel). Meanwhile, Pfs25 expression is detected on the surface of macrogametes through ookinetes with a potential role in fertilization and oocyst formation [185,186]. A number of Pfs25 fragments have been expressed using different systems including yeast, plant and viruses [179]. Initial trials of Pfs25 with Alhydrogel failed to generate robust titers redirecting testing efforts to alternative adjuvants [187,188,189].

Another arm of the transmission-blocking vaccine strategy is aimed at developing antibodies against antigens expressed in the mosquito midgut. One such antigen that has been recommended for clinical trial is a midgut-specific *Anopheles* alanyl aminopeptidase N (AnAPN1) [190,191,192]. Membrane-feeding assays have demonstrated the transmission-blocking activity of recombinant AnAPN1 fragment conjugated with Alhydrogel, underscoring its role in ookinete invasion [193,194]. These vaccines raise ethical concerns as they do not directly benefit the recipient, even though they contribute to the greater goal of building herd immunity and reducing the risk of transmission. Development of transmission-blocking vaccines is hindered by additional practical knowledge gaps, such as the lack of information on antibody titers ingested by a mosquito vector required for successful blockade [53]. Furthermore, booster doses through repeat immunizations become essential for continual stimulation of the immune system to overcome short-lasting antibody response from brief exposure of antigens. More importantly, current TBVs rely on hampering transmission at the vector-level and do not directly act to contain the infectious reservoir by reducing host gametocyte carriage.

An alternative strategy for transmission-blocking vaccines could be to develop antibodies against antigens expressed on the surface of gametocyte infected red blood cells (giRBC) [195,196,197,198]. A recent proteomic study identified about ~43 new giRBC antigens by probing human plasma samples from a malaria-endemic region with an array of gametocyte-related proteins [178]. Thirteen antigens were shortlisted based on enriched expression in gametocytes and the presence of a transmembrane domain or a secretory signal-sequence. These include antigen candidates from multigene families like surface-associated interspersed protein (SURFIN), the subtelomeric variable open reading frame (STEVOR), and type A and B repetitive interspersed family (RIFIN). These antigen candidates have not yet been characterized in gametocytes, except for STEVORs, which have been linked to gametocyte sequestration [107]. Discovery of gametocyte surface antigens have broadened the horizons of antimalarial therapeutic research; however, transmission-blocking activity of antibodies specific to these candidate antigens have not been confirmed, and they necessitate further experimental validation. New age ‘omics’ technologies can be used to identify stage-specific antigens for the development of efficacious multi-unit vaccines that will target different stages of parasite development.

## 5. Identifying Gametocyte Essential Genes in the Age of Functional Genomics

The first genome-wide transcriptome and proteome analysis of male and female *P. falciparum* gametocytes was accomplished in 2016, about fourteen years after the *P. falciparum* genome was sequenced [112]. Improved methods for in vitro gametocyte culture have made it possible to yield synchronous sexual parasite populations for such large-scale genomic studies [199,200,201]. Until then, molecular events important for gametocytes were inferred from the rodent-malaria model, including the first proteome analysis of gametocytes in *P. berghei* [202]. The fundamental differences in developmental time and morphology of gametocytes in the two species thwarts the identification of genes involved in shape-shifting, cytoskeletal rearrangements and bone marrow sequestration unique to *P. falciparum* gametocyte development. For these reasons, applications of ‘omics’ tools in *P. falciparum* have peaked in the last two decades. One of the earliest acknowledged gametocyte essential genes was linked to a deletion on chromosome 9 as a consequence of continuous in vitro culturing [203,204]. Later, the associated gene was designated as *P. falciparum gene implicated in gametocytogenesis* (*pfgig*), with a phenotype of reduced gametocyte production that could be rescued by complementation [205].

An interesting landmark was the transcriptome profiling of stage-specific (I-V) *P. falciparum* gametocytes using an expression microarray approach that led to the identification of a sexual development cluster [206]. Overall, 75% of genes from this cluster were categorized as hypotheticals, emphasizing the need for further functional annotation of genes in *P. falciparum.* Another genome-wide study conducted a comparative gene-expression analysis between the wild-type 3D7 clone and its gametocyte-deficient derivative clone F12, adding two novel genes *pfpeg-3* and *pfpeg-4* to the list of early gametocyte genes [207]. Identification of a transcriptional switch critical for *P. falciparum* sexual commitment was a milestone in shedding light on gene regulation and epigenetic mechanisms during sexual development. Forward and reverse genetics confirmed the DNA-binding protein PfAP2-G as a master regulator of sexual differentiation with early-gametocyte genes as downstream targets, including well-known markers of sexual commitment such as *pfs16, pfg27/25,* and *pfg14.74* [25,208]. Additional studies have identified gametocyte development 1 (GDV1) as an upstream activator of sexual commitment [63,64]. In an effort to further decode the network of sexual commitment, single-cell RNA sequencing (scRNA-seq) was carried out for the first time in *P. falciparum* [209]. A comparative analysis of individual parasite transcriptomes from an AP2G knockdown-line and wild-type NF54 reported an increased expression of epigenetic regulators in sexually committed schizonts, emphasizing the role of transcription factors and chromatin-modulators in sexual development [210]. A new time-course transcriptomic profiling of gametocytes was undertaken in *P. falciparum* to provide high-resolution gene expression profiles during gametocytogenesis [211].

Traditional methods of studying gene function by gene-editing have been implemented in *P. falciparum* with varying degrees of success. Recently, CRISPR-Cas9 based system was employed to assess the MAP kinase function in *P. falciparum,* where PfMAP-2 was shown to play a role in male gametogenesis [212]. Despite the significant progress that has been made in unraveling parasite biology, targeted gene-disruption approaches are time consuming and laborious. A substantial number of genes (>30%) still lack functional annotation in *P. falciparum,* underscoring the need for scalable genetic methods for functional profiling [213]. The first genome-wide genetic screen of an apicomplexan was successfully carried out in the opportunistic human pathogen *Toxoplasma gondii* that led to the identification of essential apicomplexan genes [214,215]. Meanwhile, the feasibility of large-scale reverse genetic screens was demonstrated in the rodent-malaria parasite model [216]. Follow-up studies on the use of uniquely barcoded knockout vectors in *P. berghei* resulted in the first functional screen of the *Plasmodium* genome, highlighting an abundance of essential genes for blood-stage growth [217]. More recently, a CRISPR–Cas9 based screen identified a Myb-like transcription factor (BFD1) as the master regulator of chronic-stage differentiation in *T. gondii* [218]. Undoubtedly, CRISPR-Cas9 has accelerated functional genomic studies but its application to genome-wide screens in *P. falciparum* remain restricted by the absence of nonhomologous end joining machinery and poor transfection efficiency [219,220].

## 6. Application of Forward Genetic Screens Using *piggyBac* Transposon Mutagenesis in Malaria Transmission

A long-standing strategic hurdle for genetic manipulations in *P. falciparum* has been its AT-rich (>80%) genome and the dearth of nucleotide diversity for suitable gRNA design critical to gene knock-out studies. The peculiarity of extreme AT-bias was exploited by our group for insertional mutagenesis using *piggyBac* transposons [221]. Originally isolated from the cabbage looper moth, these jumping DNA-elements preferentially insert at ‘TTAA’ target-sites using a cut-and-paste mechanism [222]. Multiple features like high efficiency of insertion, ability to generate stable mutants and cargo-delivery of considerable size (9–12 kb) make *piggyBac* transposons ideal for genetic modifications [223]. Previously developed for functional characterization of loss-of-function mutants, *piggyBac* mutagenesis was recently applied to the *P. falciparum* genome at the level of saturation, generating >38,000 single-insertion mutants [224]. An improved transfection protocol coupled with an Illumina-based sequencing method, quantitative insertion-site sequencing (QIseq) allowed the generation and tracking of random disruptions in ORFs and non-coding regions across all 14 chromosomes [224,225].

As the first whole-genome functional screen in *P. falciparum*, genes were categorized as either essential or dispensable for asexual blood-stage growth under ideal in vitro culture conditions. Two scores, mutagenesis index score (MIS), an indicator of gene-mutability, and mutagenesis fitness score (MFS), an indicator of relative fitness, were assigned to each gene to derive a reliable growth phenotype. Genes essential for blood-stage growth have a low MIS while dispensable genes rank at the top with a higher MIS (Figure 2). Notably, about 60% of the genes were determined to be essential for parasite survival, including drug resistance markers such as the Kelch propeller (*K13*), *dhfr-ts* and *mdr* [224]. In contrast, genes associated with virulence factors and other stages of parasite development like transmission and sporogony were likely to be dispensable during asexual development. A closer look at stage-wise expression reveals that about 41% (>800) of dispensable genes are highly expressed in gametocytes and thus may be important for sexual development. Further analyses confirm leading transmission-blocking vaccine candidates and master regulators of sexual commitment are dispensable for blood-stage growth (Figure 2). Large-scale phenotypic screens using *piggyBac* mutants have been recently used to identify genetic factors linked to parasite survival in heat-shock during malarial fever [226,227]. A similar approach for an ongoing phenotypic drug-screen will decipher drug-gene interactions and elucidate mechanisms of action of leading antimalarial drugs [228]. Previously, an attempt to create a gametocyte-essential gene repertoire was carried out by using *piggyBac* transposon-mediated insertional mutagenesis elsewhere [229]. The authors successfully isolated 29 gametocyte-deficient mutants that failed to form mature gametocytes. Additional analyses revealed 16 new genes that had not been previously implicated in gametocytogenesis, but confirmation of phenotype by genetic complementation was limited to five candidate genes [229]. Transposon mutagenesis offers an ideal platform for discovery of gametocyte essential genes as sexual-stage genes tend to be dispensable during asexual-growth allowing the maintenance and culture of parasites necessary for these studies. A whole genome gametocyte screen would be a worthwhile investment for unraveling genetic determinants of sexual differentiation and transmission of the deadly malaria parasite *Plasmodium falciparum*.

## 7. Concluding Remarks

Malaria remains one of the leading causes of human morbidity and mortality. As of yet, we lack a highly efficacious vaccine and are faced with an increasing threat of parasite drug resistance to artemisinin. There is an urgent need to hasten our efforts towards the discovery of novel antimalarial therapeutics and other alternate strategies to effectively contain the disease. While *P. falciparum* gametocytes can be prime targets for malaria intervention, the molecular mechanisms governing sexual development, and in turn transmission, remain largely unknown. A prospective transmission-blocking approach might include a combination-therapy of fast-acting schizontocidal drugs and long-lasting gametocidal agents to clear asexual parasites and inhibit the maturation of gametocytes, respectively. In addition, vaccinating individuals for sexual-stage antigens that target mature gametocytes would enable a reduction in gametocyte carriage of the host and assist with the prevention of local malaria transmission. The accomplishment of these goals will require assimilation of current knowledge and forthcoming data from high-throughput genomic and proteomic approaches to unlock new potential candidates. Our previous studies based on random *piggyBac* integration in the parasite genome offer a collection of mutants as an invaluable forward-genetic tool to gain unbiased and fundamental insights into parasite biology and expedite the identification and prioritization of ‘high-value’ essential transmission-blocking targets.

## Figures and Tables

**Figure 1 pathogens-10-00346-f001:**
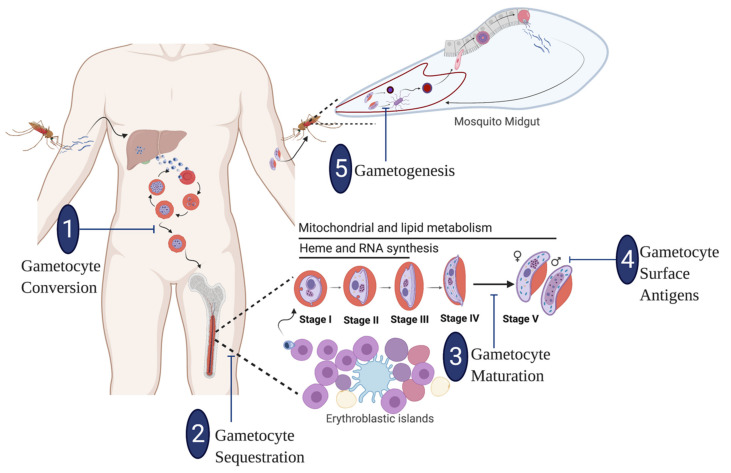
Targeting transmission stages of the malaria parasite *P. falciparum:* A schematic overview highlights various points in the parasite’s life cycle where sexual development can be interrupted by transmission-blocking agents. (**1**) Key molecular players and epigenetic regulators of sexual commitment (**2**) Facilitators of extravasation, association with erythroblastic islands and intravasation of developing gametocytes in the bone marrow (**3**) Regulators and signal transducers of gametocyte deformation (**4**) Gametocyte surface antigens for new TBVs (**5**) Sex ratio dynamics and exflagellation of male gametocytes. (The sketch was created using BioRender.com).

**Figure 2 pathogens-10-00346-f002:**
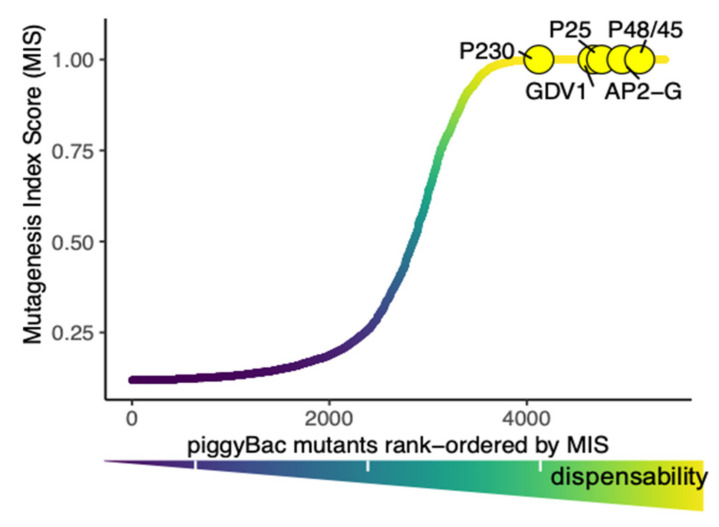
Gametocyte essential genes including the leading transmission-blocking vaccine candidates Pfs230, Pfs25, Pfs48/45 and regulators of sexual commitment such as GDV1 and AP2-G are dispensable for blood-stage growth in *P. falciparum*, suggesting the likely importance of dispensable genes in sexual development and transmission.

## Data Availability

Not applicable.

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
