# Peer review of "Targeting Gametocytes of the Malaria Parasite Plasmodium falciparum in a Functional Genomics Era: Next Steps"

_pathogens, 2021, doi:10.3390/pathogens10030346_

Round 1
Reviewer 1 Report
This manuscript excellently summarized the latest knowledge about Pf gametocytes and added useful insights from their own high-quality work (piggyBac mutagenesis).
I have only minor comments:
1) Title
Most of the description (except for transmission-blocking vaccine section) are focused on the gametocytes. But the current title "Targeting transmission stages..." sounds more general.
So, I would recommend the Authors to include "gametocyte(s)" in the Title.
2) Line 35
"Following a single cycle of silent asexual replication in the liver" should be "Following a single cycle of silent asexual replication in the liver"
3) Line 121
"a scattered nuclear pigment" should be "a scattered malaria pigment"
4) Line 229
"In contrast to the blood stages" should be "In contrast to the asexual-blood stages"
5) Line 304
"Pfs25 and pfs230 were identified as the first male and female markers, respectively" should be "Pfs230 and pfs25 were identified as the first male and female markers, respectively"
Author Response
Responses to reviewer #1:
1) Title
Most of the description (except for transmission-blocking vaccine section) are focused on the gametocytes. But the current title "Targeting transmission stages..." sounds more general.
So, I would recommend the Authors to include "gametocyte(s)" in the Title.
Thank you for your valuable comment and providing an alternative. We agree that transmission stages may not be a true indicator of human transmission stages (gametocytes) only and the term can be slightly misleading. Hence, taking the reviewer’s suggestion, we have changed our title to include gametocytes as a replacement for transmission stages. We hope this alteration justifies the content of the review and prevents any further confusion.
2) Line 35
"Following a single cycle of silent asexual replication in the liver" should be "Following a single cycle of silent asexual replication in the liver"
We have agreed to omit the use of “single cycle of” in our description of the liver stage.
3) Line 121
"a scattered nuclear pigment" should be "a scattered malaria pigment"
We accept the change of nuclear pigment to a more specific adjective- malaria pigment.
4) Line 229
"In contrast to the blood stages" should be "In contrast to the asexual-blood stages"
We agree and accept this suggestion, as blood stages could mean gametocytes as well and not specifically asexually-replicating parasites.
5) Line 304
"Pfs25 and pfs230 were identified as the first male and female markers, respectively" should be "Pfs230 and pfs25 were identified as the first male and female markers, respectively"
We have accepted and made the necessary adjustments with regards to the order of the two candidate markers.
Reviewer 2 Report
This is a timely and outstandingly complete review. Well written, comprehensive, and makes a cogent argument for biology driving potential new interventions.
Author Response
Thank you for the comments! Your positive evaluation is greatly appreciated.